# Correction of 2*π* Phase Jumps for Silicon Photonic Sensors Based on Mach Zehnder Interferometers with Application in Gas and Biosensing

**DOI:** 10.3390/s24051712

**Published:** 2024-03-06

**Authors:** Loic Laplatine, Sonia Messaoudene, Nicolas Gaignebet, Cyril Herrier, Thierry Livache

**Affiliations:** 1Univ. Grenoble Alpes, CEA, LETI, 38054 Grenoble, France; sonia.messaoudene@cea.fr (S.M.); gaignebet@insa-toulouse.fr (N.G.); 2Aryballe, 38000 Grenoble, France; c.herrier@aryballe.com; 3Univ. Grenoble Alpes, CEA, CNRS, Grenoble INP, IRIG, SyMMES, 38000 Grenoble, France; thierry.livache@cea.fr

**Keywords:** silicon photonics, Mach Zehnder interferometers, phase unwrapping, biosensors, olfactory sensors

## Abstract

Silicon photonic sensors based on Mach Zehnder Interferometers (MZIs) have applications spanning from biological and olfactory sensors to temperature and ultrasound sensors. Although a coherent detection scheme can solve the issues of sensitivity fading and ambiguity in phase direction, the measured phase remains 2π periodic. This implies that the acquisition frequency should ensure a phase shift lower than π between each measurement point to prevent 2π phase jumps. Here, we describe and experimentally characterize two methods based on reference MZIs with lower sensitivities to alleviate this drawback. These solutions improve the measurement robustness and allow the lowering of the acquisition frequency. The first method is based on the phase derivative sign comparison. When a discrepancy is detected, the reference MZI is used to choose whether 2π should be added or removed from the nominal MZI. It can correct 2π phase jumps regardless of the sensitivity ratio, so that a single reference MZI can be used to correct multiple nominal MZIs. This first method relaxes the acquisition frequency requirement by a factor of almost two. However, it cannot correct phase jumps of 4π, 6π or higher between two measurement points. The second method is based on the comparison between the measured phase from the nominal MZI and the phase expected from the reference MZI. It can correct multiple 2π phase jumps but requires at least one reference MZI per biofunctionalization. It will also constrain the corrected phase to lie in a limited interval of [−π, +π] around the expected value, and might fail to correct phase shifts above a few tens of radians depending on the disparity of the nominal sensors responses. Nonetheless, for phase shift lower than typically 20 radians, this method allows the lowering of the acquisition frequency almost arbitrarily.

## 1. Introduction

Silicon photonics make it possible to engineer highly sensitive and miniaturized sensors with high yield and high reproducibility. Among the various transduction components available, such as micro-ring resonators, photonic crystals or Bragg gratings, Mach Zehnder Interferometers (MZIs) have been widely explored and optimized [1]. MZIs are probably the easiest to design and manufacture, as they do not necessarily require a feature size below 300 nm, which is accessible using 248-UV dry lithography. They also exhibit state-of-the-art performances while only requiring commercially available optoelectronics parts, such as VCSELs for light source and CMOS imagers for light detection.

The basis of refractive-index-based sensing with integrated MZIs was introduced in the late 1980s [2] with proofs of concept for gas and biosensors in the early 1990s [3,4]. Since then, one of the major improvements was to use a coherent detection scheme to solve the issues of absolute intensity noise, sensitivity fading near extrema and ambiguity in phase direction [5,6]. Nowadays, integrated MZI sensors have been applied to a wide range of applications such as biosensing [7], temperature monitoring [8], gas detection in the ppb range [9], odor identification in the ppm range [10], ultrasound recording [11], refractive index sensing [12] down the the 10^−8^ RIU range and wavelength tracking [13], just to name a few.

Typically, the measured light intensity is converted into a phase φ expressed in radians. φ corresponds to the phase delay difference between the sensing and reference arm and its value lies in the interval [0, 2π] or [−π, π]. Thus, the signal measured by interferometric methods is always intrinsically 2π periodic. To track the physical value of interest over time, also called the sensorgram, φ needs first to be unwrapped in order to measure the accumulated phase Φ(t) which is not bonded to any interval anymore. This phase is finally converted into another unit through a calibration.

This 2π periodicity imposes a lower limit on the acquisition frequency (Nyquist frequency) to ensure that less than a π phase shift has happened between each measurement point. Otherwise, the unwrapping function will output an erroneous Φ(t), offset by multiples of 2π (referred to as “2π phase jumps” later in the text). Unfortunately, the minimum acquisition frequency is proportional to the sensor sensitivity and can reach tens of Hz for gas sensing, especially with highly concentrated samples. This implies relatively high-speed imagers and electronics, as well as powerful light sources with high optical coupling to reduce the exposure time.

Phase unwrapping error is a well-known issue in many interferometry applications, such as 3D-imaging [14], displacement sensors [15] or distributed fiber-based strain sensors [16]. However, for silicon photonic sensors, this drawback is rarely mentioned in the literature. In 2017, Milvich et al. proposed an elegant solution based on coupling a coherent detection with a wavelength modulation in order to access the absolute interference-order [17]. However, the wavelength modulation range should exceed the MZI free spectral range (FSR). Since VCSEL wavelength spanning by current modulation does not exceed a few nm, this technique requires short FSR MZIs which are more sensitive to laser wavelength drift, jitter and line-width, whereas large FSR MZIs are compatible with lower-cost lasers [18], on-chip non-monochromatic organic lasers [19] or even LEDs [20]. Moreover, low-pass filters or lock-in amplifiers are required to access the precise phase shift since several modulation periods should be averaged. The acquisition frequency should thus be tens of times faster than the typical time constant of the physical phenomenon to be tracked.

Here, we describe two different methods to detect and correct 2π phase jumps by using reference sensors with lower sensitivities. Our solution makes it possible to lower the acquisition frequency without affecting the optoelectronic hardware or the nominal MZI design. Our experimental demonstrations are based on MZI sensors but can be easily extrapolated to micro-ring resonator sensors, which suffer from the same periodicity issue with respect to their FSR.

## 2. Materials and Methods

Figure 1 show pictures of two silicon photonic dies. The left die is a typical MZI die with a 60-MZI matrix and two lower sensitivity MZIs. Each nominal MZI has a sensing arm length of 9.5 mm. The right die has been designed to assess the presented correction methods and contains 5 groups of 12 identical MZIs. The MZI arm lengths, and thus their sensitivities, are increased across the 5 groups. The sensing arm of the less sensitive group is 250 μm long. The arm lengths of the following groups are increased by factors of 7, 11, 23 and 31 to reach a maximum length of 7.75 mm. A wavelength and a temperature variation tracker are also visible in both dies but were not used in this study. The silicon photonic dies were designed and produced at CEA-Leti on a 200 mm CMOS platform as described in Ref. [10].

Measurements were performed on a custom optical bench with a single mode VCSEL emitting at λ= 850 nm aligned to the input grating coupler (GC) and a CMOS imager aligned to the output GC array. Data acquisition and processing were performed by custom MATLAB and Python scripts. Liquid samples were injected by a pressure-regulated system (Fluigent, Paris, France) at a constant flow rate. Gas samples were manually approached from the sensor input tubing while the output tubing was connected to a vacuum pump. A fluidic restriction was used to limit the flow rate. Note that the absolute values of flow rates or sample concentrations have no particular importance here since performance comparisons are relative (i.e., how much the frame rate can be reduced without signal integrity loss). Phase monitoring was performed at 20 Hz for liquid sample injection and 60 Hz for gas sample injection. The frame rate was then artificially lower in post-processing by skipping measurement points until 2π phase jumps were observed. The experiment with a liquid sample consisted in acquiring a baseline with deionized water, then injecting a Phosphate Buffer Saline solution at a concentration of 0.5× (Thermo Scientific Chemicals (Waltham, MA, USA)), and finally rinsing with deionized water. The experiment with a gas sample consisted in acquiring a baseline with air, then injecting the head space with a pure β-pinene solution (1 mL) (Sigma-Aldrich (St. Louis, MO, USA)) contained in a 10 mL flask, and finally rinsing with air.

## 3. Results

### 3.1. Phase Extraction and Phase Unwrapping Errors

Figure 2 presents the main steps to compute the phase shift ΔΦ(t) from the measured intensity variations $P_{1}$, $P_{2}$ and $P_{3}$ from a three-port coherent MZI.

Each $P_{i}$ can be written as: (1)$$P_{i}\propto 1+V\times cos\left[\varphi +\left(i-1\right)\frac{2\pi }{3}\right].$$

*V* is the visibility of the interference which varies from 1 to 0. It is a function of the lengths *L* and propagation losses α of both arms: (2)$$V=\frac{2\times e^{-\frac{1}{2}\left(\alpha_{s}L_{s}+\alpha_{r}L_{r}\right)}}{e^{-\alpha_{s}L_{s}}+e^{-\alpha_{r}L_{r}}}.$$

In Equation (Equation 1), the MZI phase φ is a function of the wavelength, the waveguide lengths and the effective indices: (3)$$\varphi =\frac{2\pi }{\lambda }\left(L_{s}\times n_{eff-s}-L_{r}\times n_{eff-r}\right).$$

From the three $P_{i}$ which are phase shifted by 2π/3, it is possible to calculate two values phase shifted by π/2 called the in-phase (*I*) and quadrature (*Q*) components: (4)$$I=P_{2}-\frac{P_{1}+P_{3}}{2},$$
(5)$$Q=\sqrt{3}\times \frac{P_{1}-P_{3}}{2}.$$

*I* and *Q* can be represented as the x and y coordinates of a point circularly turning in the IQ diagram. Alternatively, they correspond to the real and imaginary part of a complex number. In that case, the IQ diagram is equivalent to the complex plane. The phase φ corresponds to the angle of each point with respect to the x-axis: (6)φ=arctan(I,Q)=Arg(I+iQ)

It is obvious from the 4-quadrant inverse tangent function that φ can only have values from −π to +π. To access a meaningful measurement, φ needs to be unwrapped to obtain Φ as shown in Figure 2d. The phase unwrapping algorithm finds the smallest angle between two successive times *t* in the IQ diagram and adds it to the accumulated phase Φ with a positive sign if the smallest angle is found in the counter clockwise direction or a negative sign if it is found in the clockwise direction. The measured phase φ and the unwrapped phase Φ are thus always equal modulo 2π: (7)Φ=U(φ)=φ+2πn;n∈Z,
where *U* is the unwrapping function. For gas and biosensors based on long sensing waveguides, the initial phases of a matrix of MZIs are often randomly distributed because of small fabrication variations, surface biofunctionalization differences (e.g., different probe molecules on each MZI) as well as differences in surface interaction from previous experiments. Therefore, only the phase shift ΔΦ with respect to the initial phase of a measurement is taken into account: (8)ΔΦ(t)=Φ(t)−Φ(t=0).

From the phase measurement and a calibration, it is finally possible to extract the physical value of interest (temperature, wavelength, mechanical deformation or constraint, sample refractive index, surface adsorption, etc. …).

A phase unwrapping error is illustrated in Figure 3 by comparing two MZIs with different sensitivities. The phase φ of the less sensitive MZI, in blue, can be correctly unwrapped since the phase steps between two successive times *t* remain well below π. In contrast, the unwrapping function fails to output the real phase shift evolution for the most sensitive MZI, in red, as the phase step reaches a value higher than π near 9 s. This condition can be written as: (9)$$max\left\{|\frac{\partial \Delta \Phi (t)}{\partial t}|\right\}\leq \pi .$$

We can derive two different methods to detect and correct phase unwrapping errors. Both methods are based on at least one reference MZI sensor with a lower sensitivity than the nominal MZI to be corrected. It is assumed that no 2π phase jump has happened in the reference MZI.

### 3.2. First Method: Phase Shift Derivative Sign Comparison

As illustrated in Figure 4, in the first method, 2π phase jumps are detected for every time *t* when two conditions are met:The nominal and reference MZI phase shift derivatives have opposite signs:
(10)$$sgn\left(\frac{\partial \Delta \Phi_{nom}\left(t\right)}{\partial t}\right)\neq sgn\left(\frac{\partial \Delta \Phi_{ref}\left(t\right)}{\partial t}\right).$$The phase shift derivative of the nominal MZI exceeds a threshold value ε (for instance 0.3 rad). This aims at avoiding random sign noise from the flat parts of the sensorgram.
(11)$$|\frac{\partial \Delta \Phi_{nom}\left(t\right)}{\partial t}|\geq \varepsilon .$$

To obtain the corrected phase shift ${\hat{\Delta \Phi }}_{nom}\left(t\right)$, a 2π offset is added to or removed from any erroneous point according to the sign of the reference MZI derivative: (12)$${\hat{\Delta \Phi }}_{nom}\left(t\right)=\Delta \Phi_{nom}\left(t\right)+sgn\left(\frac{\partial \Delta \Phi_{ref}\left(t\right)}{\partial t}\right)\times 2\pi .$$

### 3.3. Second Method: Phase Shift Ratio Comparison

As shown in Figure 5, in the second method, the phase shift ratio *R* between the nominal and reference MZIs is used to detect erroneous points. In the absence of a phase unwrapping error, this ratio should be equal to the sensitivity ratio of the nominal ($S_{nom}$) to the reference ($S_{ref}$) MZI. *R* can be set by design, for instance, using a sensing waveguide length difference between the nominal and reference MZI, or measured experimentally during a calibration protocol as the ratio of maximum phase shifts. In the latter case, the protocol should ensure that no 2π phase jump can happen. This second method assumes that *R* is known and remains constant: (13)$$R=\frac{S_{nom}}{S_{ref}}\;or\;R=\frac{max(\Delta \Phi_{nom})}{max(\Delta \Phi_{ref})}.$$

The algorithm minimizes the discrepancy δ(t) between an expected and a potential phase shift. The former is equal to the reference MZI phase shift times the sensitivity ratio: (14)$$\Delta \Phi_{nom-expected}\left(t\right)=R\times \Delta \Phi_{ref}\left(t\right).$$

The later corresponds to the nominal MZI phase shift plus a given number of 2π phase jumps: (15)$$\Delta \Phi_{nom-k}\left(t\right)=\Delta \Phi_{nom}\left(t\right)+2k\pi \;;\;k\in Z.$$

In practice, only a limited number of 2π phase jumps can be tested at every time *t*, for instance −K≤k≤K with K=8. The smallest discrepancy δ can be written as: (16)$$\delta \left(t\right)=\min_{-K\leq k\leq K}\left\{\Delta \Phi_{nom-k}\left(t\right)-\Delta \Phi_{nom-expected}\left(t\right)\right\}.$$

Note that |δ|≤π. At every time *t*, there is a corresponding discrepancy δ(t) and a best 2π phase jump number $\hat{k}\left(t\right)$ so that the corrected phase shift can be written as:(17)$${\hat{\Delta \Phi }}_{nom}\left(t\right)=\Delta \Phi_{nom}\left(t\right)+2\hat{k}\left(t\right)\pi =R\times \Delta \Phi_{ref}\left(t\right)+\delta \left(t\right).$$

Alternatively, this method aims at finding the *k* value that makes the measured phase shift ratio $R_{k}\left(t\right)=\Delta \Phi_{nom-k}\left(t\right)/\Delta \Phi_{ref}\left(t\right)$ approach the expected one *R*.

## 4. Discussion

As seen in Figure 4a,b, the first method can efficiently correct successive phase unwrapping errors. Note that the sign discrepancy count increases as the sensitivity of the reference MZI decreases. This can be attributed to the lower signal-to-noise ratio of the reference MZI. Depending on the measurement noise floor and sensitivity ratio between the nominal and reference MZIs, a threshold can also be set on the reference MZI to avoid false error detection. This method cannot correct more than one 2π phase jump at a given time (Figure 4c). The threshold condition slightly reduces the maximum phase shift derivative that can be efficiently corrected to: (18)$$max\left\{|\frac{\partial \Delta \Phi_{nom}\left(t\right)}{\partial t}|\right\}\leq 2\pi -\varepsilon .\;$$

This improves the condition stated in Equation (Equation 9) by a factor of almost 2. Since no assumption is needed on the sensitivity ratio, a single reference MZI can be used for the correction of a full MZI matrix (Figure 1a), which makes this method efficient in terms of silicon real estate.

The second method is not restricted in terms of phase shift derivative and is even able to correct multiple 2π phase jumps at once, as shown in Figure 6. However, it relies on a constant sensitivity ratio, which, in practice, might slightly evolve during an experiment or across several experiments. This can be caused by uncompensated drifts, biofunctionalization aging and variability, poor synchronicity between the reference and nominal MZIs as well as sample inhomogeneity inside the fluidic chamber. Thus, the higher the ratio, the higher the uncertainty on the correction truthiness. However, as for the first method, the higher the ratio, the lower the risk of phase unwrapping errors on the reference MZI. A trade off must thus be found. As mentioned in Section 2, using sensitivity ratios corresponding to prime numbers slightly limits the risk of missing 2π phase jumps, as one jump on a reference MZI should not result exactly in two or more jumps on another reference or nominal MZI.

In addition, as opposed to the first method, for biosensing experiments, each biofunctionalization group must have its own reference MZI since this would affect the sensitivity ratio. In addition, if a single reference MZI is used to correct multiple nominal MZIs, it will force all their phase shifts to lie a in restricted interval of [+π, −π], while true values might exhibit a higher disparity, especially for the high absolute phase shifts. Figure 7 illustrates the aforementioned limitations and compares methods 1 and 2 for a β-pinene vapour injection. Here, the phase shifts of the 12 MZIs of the less sensitive group (in black) were averaged in order to lower the noise of the reference signal.

Figure 8 puts both methods in perspective by defining regions of efficacy with respect to the nominal MZI absolute phase shift and absolute phase shift derivative, each method being primarily limited by one of these two aspects. To robustify the correction algorithm, both methods can be combined and several reference MZIs with different sensitivity ratios can be used.

## 5. Conclusions

This study propose two new methods to relax the acquisition frequency requirement for silicon photonics sensors based on Mach Zehnder interferometers. Both methods are based on reference MZI sensors with lower sensitivities than the nominal sensors. A dedicated photonic MZI sensor die was designed, fabricated and used to acquire representative experimental data. Both methods are theoretically detailed and applied on datasets to illustrate their respective strengths and weaknesses. From these examples, we can conclude that the acquisition frequency can be lowered by a factor of ∼2 in most cases. We can also predict that if the maximum achievable phase shift can be known in advance and remains below a few tens of radians, then the acquisition frequency can be almost arbitrarily lowered. This work should help photonic designers to make more reliable and power-efficient sensors for a wide range of applications beyond gas and biosensing.

## 6. Patents

Patent referenced FR3129480 results from the work reported in this manuscript.

## Figures and Tables

**Figure 1 sensors-24-01712-f001:**
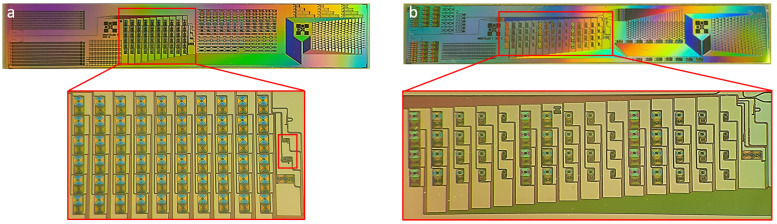
Dies with lower sensitivity reference Mach Zehnder Interferometers (MZIs). (**a**) Standard die with only two lower sensitivity reference MZIs highlighted in the red rectangle. (**b**) Special die designed to evaluate both methods with 5 sensitivity groups, each composed of 12 identical MZIs.

**Figure 2 sensors-24-01712-f002:**
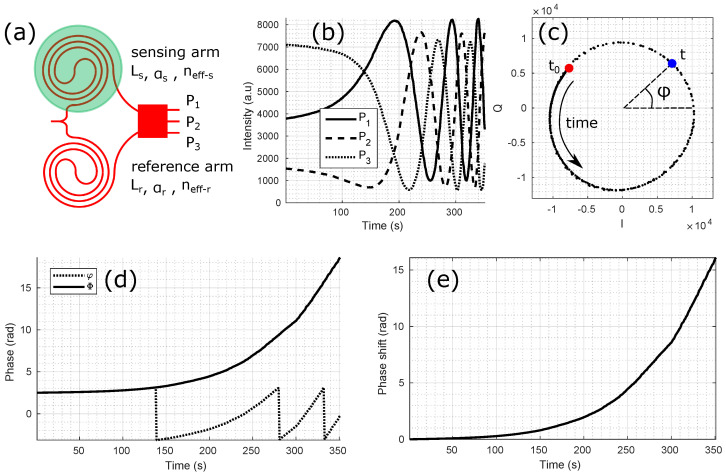
Main steps for phase shift calculation: (**a**) Schematic of a three-port MZI. (**b**) Acquisition of the three optical outputs. (**c**) I and Q diagram. (**d**) Phase unwrapping. (**e**) Phase shift.

**Figure 3 sensors-24-01712-f003:**
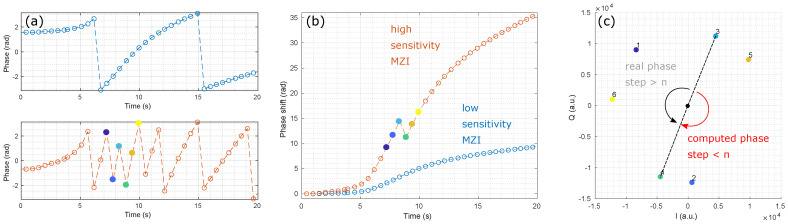
Phase unwrapping error. (**a**) Measured phase φ of two MZIs with different sensitivities. (**b**) Unwrapped phase shift ΔΦ. (**c**) IQ diagram highlighting the origin of the phase unwrapping error.

**Figure 4 sensors-24-01712-f004:**
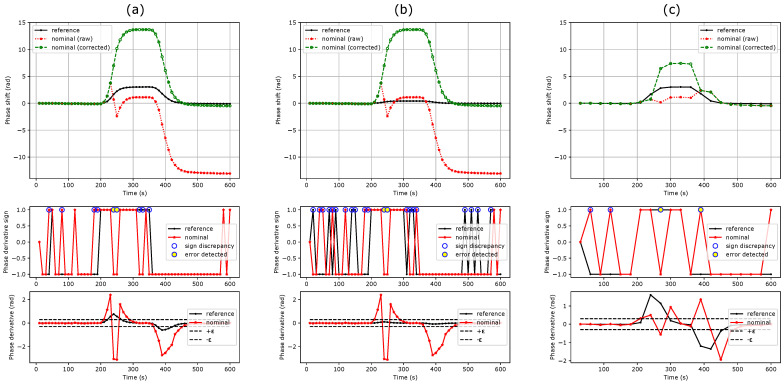
Original and corrected phase shifts using the phase shift derivative sign comparison method for a refractive index step from water to PBS 0.5×. (**a**) Sensitivity ratio of 4.5, threshold of 0.3 rad. (**b**) Sensitivity ratio of 31, threshold of 0.3 rad. (**c**) Sensitivity ratio of 4.5, threshold of 0.3 rad at a frame rate divided by 3. This method fails to retrieve the real phase shift.

**Figure 5 sensors-24-01712-f005:**
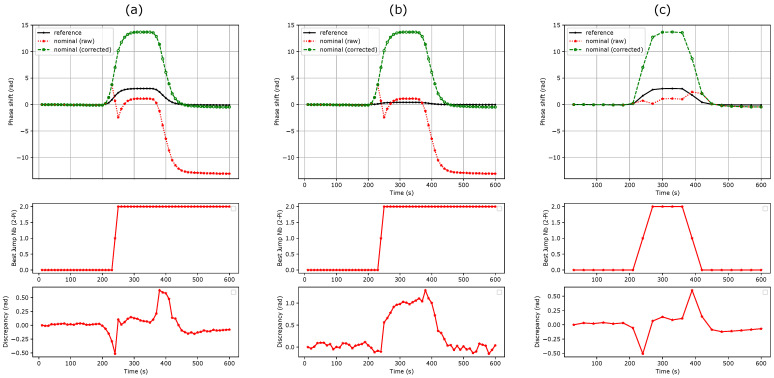
Original and corrected phase shifts using the phase shift ratio comparison method. (**a**) R=4.5. (**b**) R=31. (**c**) R=4.5 at a frame rate divided by 3. The method remains efficient regardless the frame rate reduction.

**Figure 6 sensors-24-01712-f006:**
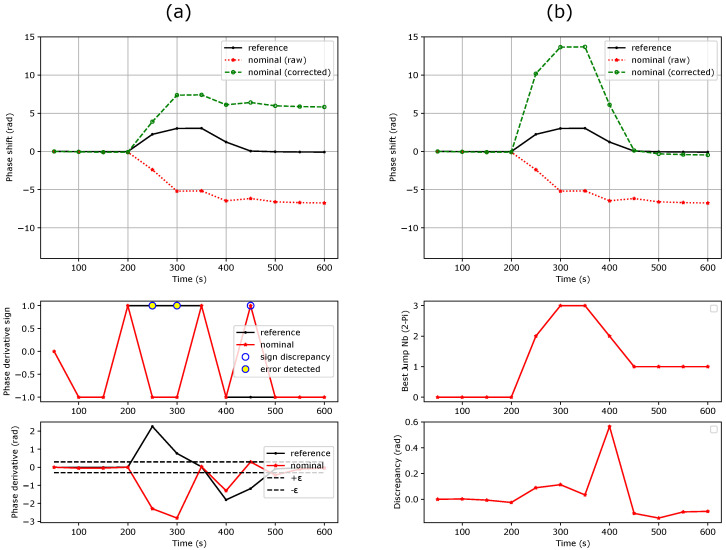
Original and corrected phase shifts using both methods in the case of multiple 2π phase jumps (frame rate further reduced by a factor of 1.7 compared to Figure 4 and Figure 5). (**a**) Method 1 fails. (**b**) Method 2 succeeds.

**Figure 7 sensors-24-01712-f007:**
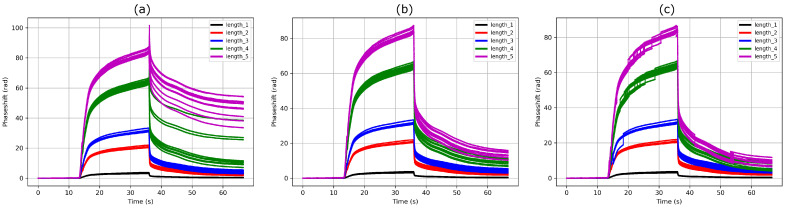
β-pinene vapor injection on die b. The 5 different sensitivity groups are clearly visible, each containing 12 identical MZIs. The mean of the less sensitive group is used as reference. (**a**) Phase unwrapping errors happen for the two most sensitive groups during desorption in the original phase shift. (**b**) Method 1 succeeds in correcting these errors. (**c**) Method 2 fails at perfectly correcting the two most sensitive groups and generates new errors in the three most sensitive groups.

**Figure 8 sensors-24-01712-f008:**
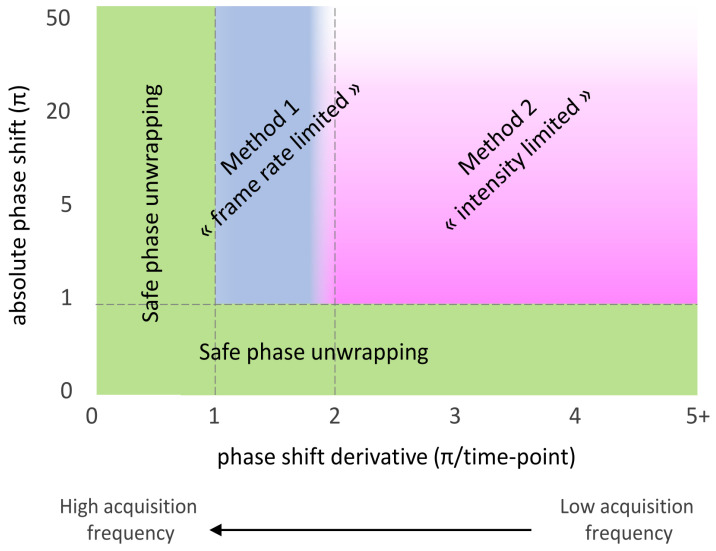
Region of efficacy of the presented methods in terms of absolute phase shift (i.e., intensity) and absolute phase shift derivative (i.e., frame rate).

## Data Availability

Dataset available on request from the authors.

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
