# Peer review of "Correction of 2π Phase Jumps for Silicon Photonic Sensors Based on Mach Zehnder Interferometers with Application in Gas and Biosensing"

_sensors, 2024, doi:10.3390/s24051712_

Round 1

Reviewer 1 Report

Comments and Suggestions for Authors

The present study is interesting and important and the obtained results are significant. Hence, the manuscript can be accepted in its current form.

Author Response

Dear Reviewer, 

Thank you very much for taking the time to review our work. 

Reviewer 2 Report

Comments and Suggestions for Authors

Please review the document.

Comments on the Quality of English Language

The English is fluent to read, but the description of the experimental content needs to be more detailed.

Author Response

Dear Reviewer,

Thank you very much for taking the time to review this manuscript. Please find the detailed responses below and the corresponding revisions highlighted in the re-submitted file.

Point-by-point response to Comments and Suggestions for Authors :

  • Comment 1: Abstract. The experimental principles and the significance of the research can be written in the abstract, which can help capture the readers' interest.

Response:  The abstract was indeed very short. We have added the following text at the end of the original abstract to address your comment:

"The first method is based on the phase derivative sign comparison. When a discrepancy is detected, the reference MZI is used to choose whether 2pi should be added or removed from the nominal MZI. It can correct 2pi phase jumps regardless of the sensitivity ratio, so that a single reference MZI can be used to correct multiple nominal MZI. This first method relaxes the acquisition frequency requirement by a factor of almost two. However, it cannot correct phase jumps of 4pi, 6pi or higher between two measurement points. The second method is based on the comparison between the measured phase from the nominal MZI and the phase expected from the reference MZI. It can correct multiple 2pi phase jumps but requires at least one reference MZI per biofunctionalization. It will also constrain the corrected phase to lie in a limited interval of [-pi,+pi] around the expected value, and might fail to correct phase shifts above a few tens of radians depending on the disparity of the nominal sensors responses. Nonetheless, for phase shift lower than typically 20 radians, this method allows lowering the acquisition frequency almost arbitrarily."

  • Comment 2: "Keywords. The keyword can be added, such as phase unwrapping."

Response:  We have added the keyword “phase unwrapping”.

  • Comment 3: "Line 63. It is suggested to provide a detailed description of the design method for the sepcial die in Figure 1b. Moreover, does Figure 1b depict 15 groups of 4 MZI?"

Response:  We are very grateful you pointed out a mistake in the original manuscript. There are not 15 groups of 5 MZI, but rather 12 groups of 5 MZI. And after considering it, it also appears clearer to state that there are 5 groups of 12 MZI. We emphasized that the MZI length increases across these 5 groups, leading to 5 sensitivity groups, and that each group is thus composed of 12 identical MZI.  We also mentioned the lengths and length ratios. The fact that the ratios are prime numbers is discussed in the discussion section line 193. The length of the MZI in Figure 1a has also be added.

  • Comment 4: "Line 69, It is better to supplement an experimental flowchart so that the reader can understand the experimental procedure intuitively"

Response:  Both the liquid and gas experiments are quite similar and are just here to exemplify the strengths and weaknesses of our correction methods. Therefore, we believe adding a new figure is not necessary and we would prefer to address this point by describing the protocol for each experiment. We also added the supplier for the chemicals. We have added the following text at the end of the material and method section (line 96 to 101):

"The experiment with a liquid sample shown in Figure 4, 5 and 6 consisted in acquiring a baseline with deionized water, then injecting a Phosphate Buffer Saline solution at a concentration of 0.5X (Thermo Scientific Chemicals), and finally rinsing with deionized water. The experiment with a gas sample shown in Figure 7 consisted in acquiring a baseline with air, then injecting the head space of a pure β-pinene solution (1 ml) (Sigma-Aldrich) contained in a 10 ml flask, and finally rinsing with air."

  • Comment 5: "Line 117. An explanation for Figure 3 can be added, comparing the phase variation of MZIs with different sensitivities."

Response:  Again, we are very grateful you pointed out this figure, as there was a mistake in the original manuscript. The absolute phase of the low sensitivity MZI, in blue, in Figure3a does not correspond to the unwrapped one in Figure 3b. The absolute phase was the one from another nominal MZI with nearly the same sensitivity as the one shown in red (which undergo a phase unwrapping error). We corrected Figure3a with the correct absolute phase from the reference MZI. This makes the figure easier to understand. We have nonetheless added the following explanation for Figure3 (line 135-139):

“A phase unwrapping error is illustrated in Figure 3 by comparing two MZI with different sensitivities. The phase φ of the less sensitive MZI, in blue, can be correctly unwrapped since the phase steps between two successive time-points t remain well below π. In contrast, the unwrapping function fails to output the real phase shift evolution for the most sensitive MZI, in red, as the phase step reaches a value higher than π near 9 s.”

  • Comment 5: "Figure 4 and 5. It is better to introduce and analysis the charts after Figure 4 and 5."

Response:  We have repositioned the figures the best we could, but Latex sometimes imposes the figure position. They now appear closer to where they are discussed in the text.

Questions for General Evaluation

Are the methods adequately described? à Must be improved

Response : We hope the corrections and text additions we made have sufficiently improved the manuscript for publication.

Reviewer 3 Report

Comments and Suggestions for Authors

The manuscript “Correction o 2\pi phase jumps…” focuses on correction of large phase jumps (more than  2\pi) in the integrated system of Mach Zender interferometers used for sensing. The problem of cycle slips which can occur in any phase sensitive device is  important for correct interpretation of experimental results. Typically, it is solved by adding a subsystem with significantly lower phase sensitivity,  using  different types of filtering, parallel reading and many others . The authors exploit the first mentioned  approach building an additional low-phase-sensitivity MZI on the same chip. Interpretation for further correction of the phase jumps is done by two ways: (i) analyzing the derivative of the phase pattern or (ii) monitoring the ratio of the MZI’s sensitivities. Both of the methods give satisfactory results.

The manuscript is of scientific value, because authors demonstrate their approach  in experiment using specially manufactured photonic die developed in their laboratory. Multiple experiments support author’s evaluations which allow to implement the approach for really working sensors improving their dynamic range. Both scientifically and topically, the manuscript well fits the scope of “Sensors” and recommended for publication after small revisions.

For example, using “atan(I,Q)” seems to be a part of Python code, please use regular arctan(x/y). Line (9) : unit “\Phi” is not defined. I ask authors to check for such minorities through the text. Otherwise, the manuscript can be accepted as it is.

Comments on the Quality of English Language

English quality is fine

Author Response

Dear Reviewer, 

Thank you very much for taking the time to review our work. Please find our responses below and the corresponding revisions in track changes in the re-submitted file:

Comment: "using “atan(I,Q)” seems to be a part of Python code, please use regular arctan(x/y)".

Response: Atan2 was apparently developed in FORTRAN and is now sometimes used outside of FORTRAN codes. We could not find an official convention for the mathematical abbreviation of the 4-quadrant inverse tangent function. Since the arctan(x/y) has value in [-Pi/2;Pi/2] it is not strictly equivalent to atan2. Another convention is to use the notation arctan(x,y) (in Wolfram Mathematica language for instance). We decided to use this later convention. To avoid any confusion, we also mentioned its equivalence with the argument function ("arg") for complex numbers x + iy, which is also sometimes used in interferometry with coherent detection. The text describing Equation 5 (line 115-120) is now :

“I and Q can be represented as the x and y coordinates of a point circularly turning in the IQ diagram. Alternatively, they correspond to the real and imaginary part of a complex number. In that case, the IQ diagram is equivalent to the complex plane. The phase $\varphi$ corresponds to the angle of each point with respect to the x-axis:

\varphi = arctan(I,Q) = Arg(I+iQ)                            (5)

It is obvious from the 4-quadrant inverse tangent function that \varphi can only have values from -\pi to +pi.”

Comment: "Line (9): unit “\Phi” is not defined."

Response: Our line (9) corresponds to the keywords section. It seems the line numbering is not the same between our pdf file and the one used in the review process. However, if you refer to the unit of \Phi, it is the same as the unit of \phi since both phase are modulo 2-pi as stated in Equation 6. Therefore, it does not seem necessary to precise it again. 

Comment: "I ask authors to check for such minorities through the text."

Response: We went through the text to correct some typos and improve slightly the English.

We hope we have addressed your comments and improve the manuscript for publication in Sensors.

Round 2

Reviewer 2 Report

Comments and Suggestions for Authors

Please review the file.

Comments on the Quality of English Language

The English of this manuscript is fluent and easy to read.

Reviewer 3 Report

Comments and Suggestions for Authors

I thank authors for comments. The manuscript can now be published.